# Evaluating the impacts of tiered restrictions introduced in England, during October and December 2020 on COVID-19 cases: a synthetic control study

Xingna Zhang ,[1] Gwilym Owen ,[1] Mark A Green ,[2] Iain Buchan ,[1] Ben Barr [1]

[1]Department of Public Health, Policy and Systems, University of Liverpool, Liverpool, UK
[2]Department of Geography and Planning, University of Liverpool, Liverpool, UK

**Correspondence to**
Dr Xingna Zhang;
xingna.zhang@liverpool.ac.uk

## ABSTRACT

**Objectives** To analyse the impact on SARS-CoV-2 transmission of tier 3 restrictions introduced in October and December 2020 in England, compared with tier 2 restrictions. We further investigate whether these effects varied between small areas by deprivation.

**Design** Synthetic control analysis.

**Setting** We identified areas introducing tier 3 restrictions in October and December, constructed a synthetic control group of places under tier 2 restrictions and compared changes in weekly infections over a 4-week period. Using interaction analysis, we estimated whether this effect varied by deprivation and the prevalence of a new variant (B.1.1.7).

**Interventions** In both October and December, no indoor between-household mixing was permitted in either tier 2 or 3. In October, no between-household mixing was permitted in private gardens and pubs and restaurants remained open only if they served a 'substantial meal' in tier 3, while in tier 2 meeting with up to six people in private gardens were allowed and all pubs and restaurants remained open. In December, in tier 3, pubs and restaurants were closed, while in tier 2, only those serving food remained open. The differences in restrictions between tier 2 and 3 on meeting outside remained the same as in October.

**Main outcome measure** Weekly reported cases adjusted for changing case detection rates for neighbourhoods in England.

**Results** Introducing tier 3 restrictions in October and December was associated with a 14% (95% CI 10% to 19%) and 20% (95% CI 13% to 29%) reduction in infections, respectively, compared with the rates expected with tier 2 restrictions only. The effects were similar across levels of deprivation and by the prevalence of the new variant.

**Conclusions** Compared with tier 2 restrictions, additional restrictions in tier 3 areas in England had a moderate effect on transmission, which did not appear to increase socioeconomic inequalities in COVID-19 cases.

## INTRODUCTION

In Autumn 2020, England experienced a second wave of COVID-19 cases with

## Strengths and limitations of this study

► The synthetic control method for microdata offers a rigorous method for identifying control areas that experienced similar levels of transmission as the intervention areas prior to the introduction of tiered restrictions supporting a casual interpretation of the finding of reduced transmission in areas with greater restrictions following their introduction.

► The use of small area data enabled interaction analysis to estimate whether this effect varied by level of deprivation.

► Our analysis assumes that case detection rates are similar across small areas in each local authority and the relationship between infections and hospitalisations (ie, the infection hospitalisation rate) remained constant over the study period.

► If there were differential trends in case detection rates between intervention and control areas, this could bias the results.

► There may also be other differences between intervention and control areas, beyond those included in this study, that led to the differences in the trajectory of SARS-CoV-2 infections observed, such as individual or household characteristics

prevalence increasing 10-fold from 0.1% in August to 1% in October.[1] Reported cases of COVID-19 were unevenly distributed across the country, with areas in the North of England most severely affected. Initially, a variety of local restrictions were introduced, which became a standardised three-tier system in October. This was followed by a month-long national lockdown, with tiered restrictions reintroduced in December and a further national lockdown in January 2021.

During the pandemic, evidence has accumulated showing that restrictions such as closing schools, public event bans and stay-at-home orders have substantially reduced

transmission of SARS-CoV-2.[2] Most evidence is, however, based on national implementation of policies[2 3] and cross-country comparisons,[2] which may not be applicable to restrictions varying across small areas within countries. Much evidence is also based on simulation rather than empirical studies, whereby observed changes in mobility and survey-based indicators of social contact following interventions are used to predict the expected impact of restrictions on cases and hospitalisations using compartmental models.[4] Estimates of intervention impact from such studies may differ from the observed impact on cases, if actual relationships between mobility and transmission following intervention differ from simulation assumptions.

There are also concerns that such restrictions may have differential effects between different socioeconomic contexts. This is because people of disadvantaged communities may not have the same access to outdoor spaces for socialising, there may be differences in the use of restaurants and pubs and differences on the effectiveness of communications aiming to increase compliance with restrictions.[5] Understanding the potential differences in effect of control measures by socioeconomic group is important because the pandemic has disproportionally affected more disadvantaged groups.[6 7] Interventions designed to control the pandemic may further exacerbate these inequalities if we do not evaluate their differential effects and take actions to mitigate any intervention-generated inequalities.[8]

In this study, we analyse the impact on SARS-CoV-2 transmission of tier 3 restrictions introduced in October and December in England, compared with tier 2 restrictions, where the main differences were additional restrictions on meeting people outdoors and the hospitality sector in tier 3 areas. We further investigate whether these effects varied between small areas by level of deprivation.

## METHODS
### Patient and public involvement
Patients and public were not involved.

### Data and setting
We use UK government data on weekly number of people with at least one positive COVID-19 test result[9] living in 7201 Middle Layer Super Output Areas (MSOAs), between September 2020 and January 2021. MSOAs are standard geographical units used to report statistics in England, with an average population of 8000. Where there were fewer than three cases in any given week, the number of cases was supressed. In these situations, we imputed the number of cases, using complete data available at a higher geographical level (local authority (LA)), so that the sum of cases across MSOAs within a LA was equal to the total number of weekly cases reported for that LA. In total, 4% of the outcome data was imputed (see online supplemental appendix 1 for further details). LAs are municipalities covering the whole of England

and have largely been the subnational geographical units used to organise response, testing and control measures during the pandemic.

As trends in reported cases are affected by changes in testing capacity, testing strategy and public behaviour. We adjusted the weekly cases reported for each MSOA by dividing it by a weekly case detection rate estimated in each LA, calculated using a method outlined by Kulu and Dorey (see online supplemental appendix 2).[10]

We also measured local area characteristics that could potentially influence transmission and/or effectiveness of control measures. These included the overall score of the English Indices of Multiple Deprivation (IMD)—a composite measure of socioeconomic disadvantage that combines seven domains of deprivation (income, employment, education, health, crime, barriers to housing & services and living environment),[11] average number of care home beds per capita from the Care Quality Commission, population density calculated as the mid-2019 population estimates for each MSOA divided by area of the MSOA in hectares from the Office for National Statistics,[12 13] the percentage of over 70 and 7–11 population using 2019s mid-year population estimates from the Office for National Statistics, the proportion of the Black Asian and Minority Ethnic (BAME) groups and the proportion of students from the 2011 Census. To additionally account for potential differences in testing between areas, we used data for LAs on the number of tests per capita in the 4 weeks prior to the intervention available from the UK government COVID-19 dashboard. To account for differences in the prevalence of the new variant B.1.1.7,[14] we included the proportion of positive tests with S-gene failure on PCR testing for each LA from Public Health England.[15]

This time series of MSOA weekly data, area characteristics and linked LA data were then merged with a data set, indicating the restrictions that each area was under any given week. Data on restrictions were compiled and made available from the Open Data Institute.[16]

### Intervention
In tiered restrictions introduced in both October and December, no mixing between households was permitted indoors in either tier 2 or tier 3. The main difference in October was that in tier 3, people were prohibited from meeting with people outside their household in private gardens and pubs and restaurants were only allowed to remain open if they were serving a 'substantial meal', while in tier 2, people were allowed to meet with up to six people in private gardens and all pubs and restaurants remained open. In December, in tier 3 pubs and restaurants were closed, while in tier 2, only those serving food remained open. The differences in restrictions between tier 2 and 3 on meeting outside remained the same as in October. See online supplemental appendix 3 for further details.

### Analysis
We investigate two intervention time points, weeks commencing 19 October and 7 December 2020. In each

period, the initial allocation to tiers was announced on Friday and we take the week starting the following Monday as the start of the intervention. We define MSOAs as being in the intervention group if they were in tier 3 at those time points. We investigate the change in cases in the intervention group, 4 weeks before and after that time point, compared with a synthetic control group derived from places that entered tier 2 at the same time. We analyse the two groups based on their initial allocation, even though in October 22% of the MSOAs initially allocated to tier 2 were later moved to tier 3. All the MSOAs, apart from those in one LA (Kent), initially allocated to tier 3 in December stayed in tier 3 until the country entered a national lockdown beginning at January 2021. Analysing the groups based on their initial allocation is analogous to an intention to treat analysis in a trial and will provide a more conservative estimate of effect size. This will be less prone to bias of selecting places based on their subsequent transition into Tier 3, which in itself would be influenced by the effectiveness of the tiered restrictions.[17]

We apply the synthetic control method for microdata developed by Robbins *et al* to estimate the intervention effect.[18 19] The synthetic control method is a generalisation of difference-in-difference methods, whereby an untreated version of the treated cases (ie, a synthetic control) is created using a weighted combination of untreated cases. As the allocation to tier 3 areas was based on the average level and trend in cases at the LA level, this method is able to identify many small areas of tier 2 that had similar levels and trends in cases as tier 3 areas before the introduction of restrictions.

To construct the synthetic control group, we derive calibration weights to match the MSOAs in tier 2 to tier 3 areas across the 4-week period prior to the intervention by local area characteristics described earlier and the corresponding case trends and levels. For the October period, we additionally included the number of weeks each area experienced local restrictions before introducing the tiered system. The weighting algorithm derives weights that meet three constraints. First, the sum of weights in the control group equals the number of cases in the intervention group. Second, the weighted average of each local area characteristic in the synthetic control group matches those in the intervention group. Finally, the synthetic control and intervention group also match across all preintervention time points in case numbers.[18]

The Average Treatment Effect for the Treated (ATT) is estimated as the difference in cumulative number of cases in the intervention group in the 4 weeks after the intervention time point, compared with the (weighted) number of cases in the synthetic control group. To estimate the 95% CIs and p values, we apply a permutation procedure, through repeating the analysis through 250 placebo permutations randomly allocating Tier 2 MSOAs to the intervention group.[19] All analyses were performed using R V.4.0.3 and the Microsynth package.[18]

To investigate whether there was a differential effect by socioeconomic group, we grouped the MSOAs into three equal sized groups (tertiles) by level of deprivation. We then reran the weighting algorithm, stratifying the process by IMD tertile. The ATT can be estimated using the calibration weights in a weighted generalised linear model with the binary variable for the intervention group as the exposure. Accounting for the distribution of the outcome data, we fitted a weighted Poisson model with a log link function and the stratified weights, alongside an interaction term between IMD tertile and the intervention indicator.

In December, transmission and potentially the effectiveness of interventions were affected by the emergence of a new variant (B.1.1.7). We used the same approach as above for IMD to investigate differences in the effect of entering tier 3 in December, based on the prevalence of the new variant in each area.

### Sensitivity analysis

To explore sensitivity to different assumptions, we used the confirmed COVID-19 cases instead of wider case-detection rates as our outcome and we found larger effects in both time periods (see online supplemental appendix 4). When excluding the tier 2 MSOA areas located within 20 km of tier 3 areas, we found smaller effects but with high p values (see online supplemental appendix 5). This suggests that there may have been some spill-over effects, whereby travel from tier 3 areas to neighbouring tier 2 areas contributed to a rise in transmission in neighbouring tier 2 areas. However, such effects may well have occurred by chance.

### RESULTS

We created a map to show the areas that entered tier 3 at the two time points (figure 1). In October, the initial tier 3 areas (391 MSOAs) were entirely in the North West of England. In December, a larger proportion of England (2858 MSOAs) initially entered Tier 3.

We presented summary statistics for areas within each tier for both time points in table 1. As would be expected, estimated SARS-CoV-2 infection rates were higher in tier 3 areas prior to the introduction of the tiered system at both points. Tier 3 areas were more deprived on average and had a lower proportion of the population from BAME groups and lower population density. There were no differences in terms of students and care homes. In December, the new variant was more prevalent in tier 2 than tier 3 areas as measured by the proportion of S-gene target failure (SGTF) in routine PCR. In constructing the synthetic control group, weights were calculated to minimise the difference in each of the variables in table 1. As an exact match was achieved, the weighted average of each of these variables in the control group was identical to the intervention group. A map showing the geographical pattern of these weights is given in online supplemental appendix 6.

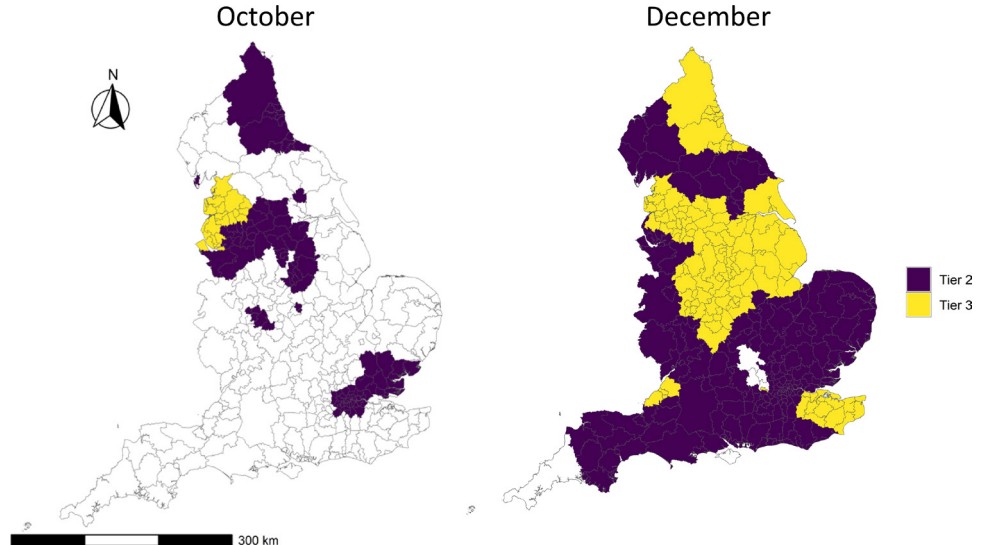

October          December

Tier 2
Tier 3

**Figure 1** Location of areas that entered tier 3 (yellow) and tier 2 (purple) at the two intervention time points.

We mapped out the trend in infection rates in the intervention and synthetic control areas before and after intervention (figure 2). As an exact match was achieved, these trends were identical in the synthetic control and intervention groups in the preintervention period. In October, the rates were increasing before the tiered system was introduced. They then started to fall, with the drop starting first in the tier 3 areas. With the second implementation of the tiered system in December, infection rates were falling while the country was in national lockdown, then increased as the national lockdown came to an end and the tiered system was reintroduced. This

increase, however, was slower in the tier 3 areas compared with the synthetic control. See online supplemental appendix 7 for charts displaying the differences between outcomes in the intervention and synthetic control groups compared with 250 placebo permutations.

We compared the estimated effect of tier 3 restrictions to what would have been expected if tier 2 restrictions had been applied on those areas (table 2). In October, the introduction of tier 3 restrictions was estimated to have led to 14% fewer cases (95% CI 10% to 19%), than what would have been the case if tier 2 restrictions would have been applied. In December, the tier t3 restrictions

**Table 1** The comparison between the tier 3 and tier 2 areas at the two intervention time points in the 4 weeks prior to the introduction of the tiered system

|  | October 2020 | | December 2020 | |
| --- | --- | --- | --- | --- |
|  | **Tier 3** | **Tier 2** | **Tier 3** | **Tier 2** |
| Average % estimated case detection rate (confirmed cases/estimated number of infections) in 4 weeks before tiers introduced | 45 | 52 | 41 | 47 |
| Average tests per 100 000 per week in 4 weeks before tiers introduced | 3232 | 2354 | 2970 | 2752 |
| Weekly infections per 100 000 per week in 4 weeks before tiers introduced | 723 | 307 | 784 | 355 |
| Care home beds per 10 000 population | 98 | 69 | 86 | 80 |
| IMD score | 31 | 25 | 26 | 19 |
| Population density—people per hectare | 32 | 51 | 30 | 41 |
| % of population 70+ | 14 | 12 | 14 | 14 |
| % population 7–11 years old | 6 | 6 | 6 | 6 |
| % BAME | 7 | 22 | 12 | 15 |
| % S-gene target failure in routine PCR | NA | NA | 23 | 41 |
| % students | 3 | 4 | 3 | 3 |
| Total population | 3 068 261 | 25 272 230 | 23 347 218 | 31 682 197 |
| Number of MSOAs | 391 | 2994 | 2858 | 3774 |

BAME, Black Asian and Minority Ethnic; IMD, Indices of Multiple Deprivation.

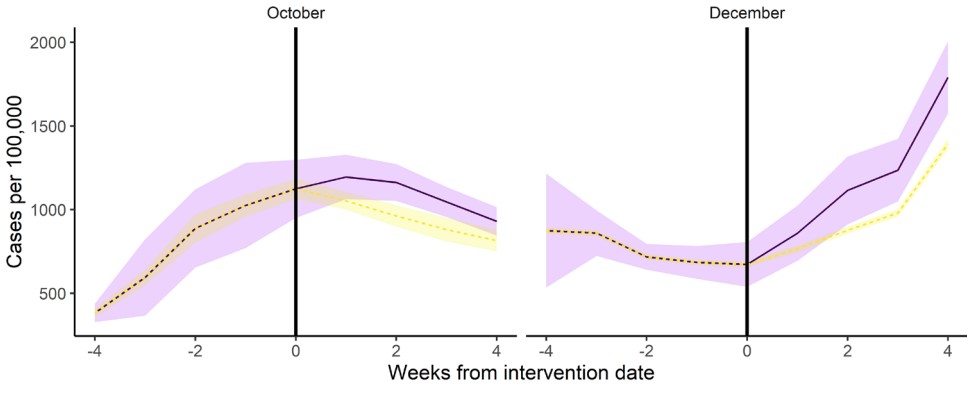

**Figure 2** The trend in case rates with their 95% CIs in the tier 3 areas and in the synthetic control group.

are estimated to have led to a slightly greater reduction in cases of 20% (95% CI 13% to 29%).

In the subgroup analysis by deprivation, there is a statistically significant effect for each level of deprivation for each time period, suggesting that tier 3 interventions had an effect across all levels of deprivation. In October, the effect estimate tended to be greater the more deprived an area was, although there was a high probability that this difference occurred by chance with high p values. In December, the effect was similar across all levels of deprivation and no interaction effects were detected again. Including an interaction term between the proportion of cases that were the new variant B.1.1.7 and the intervention group, suggested that the effect of tier 3 restrictions may have been greater in areas where the new variant was more prevalent (−27%, 95% CIs −35% to −18%), but again the p values for this interaction were greater than 0.05, indicating substantial uncertainty.

## DISCUSSION

Our study presents timely empirical evidence of the effectiveness of implementing regional tiered restrictions to manage COVID-19 responses. We find more stringent regional restrictions effective at reducing infection rates. For both time periods, areas placed in tier 3 restrictions experienced a moderate reduction in SARS-CoV-2 infections compared with what would have been expected if the same areas had been placed into tier 2. This is consistent with reporting from the UK Scientific Advisory Group on Emergencies—which concluded in November that the October tier 3 restrictions had reduced transmission, although they concluded that the effect could not be quantified and that it was unclear if tier 3 restrictions alone would be sufficient to reduce R below 1.[20] One simulation study predicted that moving into tier 3 restrictions in England in October reduced the effective

**Table 2** Results of the synthetic control analysis—indicating the relative reduction in infections in tier 3 areas compared with what would have been expected if tier 2 restrictions had been applied

| | Percentage change in cases | 95% CI | | | P value for interaction in subgroup analysis |
| --- | --- | --- | --- | --- | --- |
| | | LCL | UCL | P value | |
| October 2020—all tier 3 | −14 | −19% | −10% | <0.001 | |
| Most affluent areas | −10 | −17% | −2% | 0.016 | |
| Intermediate deprivation | −15 | −22% | −7% | <0.001 | 0.354 |
| Most deprived areas | −19 | −29% | −7% | 0.003 | 0.214 |
| December 2020—all tier 3 | −20 | −29% | −13% | <0.001 | |
| Most affluent areas | −19 | −24% | −14% | <0.001 | |
| Intermediate deprivation | −26 | −38% | −11% | 0.001 | 0.362 |
| Most deprived areas | −14 | −26% | 0% | 0.046 | 0.448 |
| Low SGTF (2%–20%) | −13 | −30% | 9% | 0.236 | |
| Intermediate SGTF (21%–44%) | −6 | −16% | 5% | 0.271 | 0.579 |
| High SGTF (45%–85%) | −27 | −35% | −18% | <0.001 | 0.174 |

Interaction analysis shows differences in effect by level of deprivation and prevalence of variant B.1.1.7 indicated by SGTF where quantitative reverse transcriptase PCR is used for COVID-19 diagnosis.
SGTF, S-gene target failure.

reproduction number (R) by 10%,[4] which is similar to the effect size we estimate here.

This study adds to the evidence that the additional restrictions on outdoor meeting and the hospitality sector have an important role to play in controlling transmission. The two main differences between tier 3 and tier 2 were that people in tier 3 were not allowed to meet with people outside their household in private gardens and there were greater restrictions on pubs and restaurants. We cannot tell from our analysis whether either additional restrictions or a combination of both were responsible for the effects we observed. There is, however, some previous evidence indicating that hospitality settings do play a role in transmission,[21–24] and outdoor proximity is thought to have a low risk of transmission.[25] We find no consistent evidence that the effects differed by levels of deprivation. While broader lockdowns that require people to work from home where possible—might be expected to have differential effects by socioeconomic group, with more disadvantaged groups being less likely to be able to work from home, this is, not necessarily the case with restrictions to the hospitality sector. It may be the case that transmission within the hospitality sector occurs at a similar level in more deprived and more affluent neighbourhoods and the restrictions introduced reduce these risks by similar amounts.

The emergence of new more infectious variants of SARS-CoV-2 raises concerns regarding the effectiveness of existing control measures. Increased infectiousness could mean that some activities such as outdoor contact previously seen as low risk could become higher risk. Our results suggest that it is plausible that in the second period of our study, when the new variant B.1.1.7 was more prevalent, the marginal effect of restrictions on outdoor meeting and hospitality settings was relatively greater. There was, however, high statistical uncertainty with this finding—possibly inflated by variant surveillance being more accurate in some areas than in others. Further investigation is needed to understand the effectiveness of these restrictions in the presence of new variants.

Our analysis has some limitations. First, we were only able to adjust for variation in the case detection rate using a relatively crude measure estimated at the LA level. This estimate assumes that the infection hospitalisation rate and the infection fatality rate do not vary between places that have similar prevalence of underlying health conditions and do not vary over the study periods. Our analysis also assumes that the case detection rate is constant across MSOAs within each LA. We do, however, find larger effects when not applying our estimated case detection rate and we also adjusted for differences in the amount of testing carried out in each area. Second, although we were able to match areas to ensure a good balance of potential confounding factors prior to the intervention, it is still possible that unmeasured variables could bias the results. Third, we were only able to use data on small neighbourhood areas, rather than on individuals and, therefore were unable to investigate how effects of control measures varied by individual or household characteristics.

As countries such as the UK continue the battle to control COVID-19 cases, with large regional differences in transmission, tiered restrictions as well as national lockdowns will likely be needed to reduce infection levels while sufficient effective vaccination coverage is achieved. At present, the UK government's plans for exiting the current lockdown are for a staged easing of restrictions at the same time across the whole of England.[26] Concerns have been raised about this—one size fits all—approach, given that high infection rates persist—particularly in more deprived areas.[27] Our analysis indicates that tiered restrictions in outdoor gathering and in the hospitality sector are effective at moderately reducing the growth of cases and could be part of an effective strategy for reducing geographical differences in transmission risk as we emerge from the pandemic.

**Contributors** B.B. is the guarantor and accepts full responsibility for the finished work and the conduct of the study. B.B. conceived the original Idea, devised the project, the main conceptual ideas and proof outline, had access to the data, and controlled the decision to publish. XZ and BB carried out the data analysis. XZ drafted the manuscript and designed the figures with support from GO and BB, GO aided in improving the model specification, and MAG aided in interpreting the results and commented on the manuscript. IB aided in supervising the project and commented on the manuscript. All authors provided critical feedback and helped to shape the research manuscript.

**Funding** This study is funded by the National Institute for Health Research (NIHR) Health Protection Research Unit in Gastrointestinal Infections, a partnership between Public Health England, the University of Liverpool and the University of Warwick. Grant Number NIHR ref NIHR200910. The views expressed are those of the author(s) and not necessarily those of the NIHR, Public Health England or the Department of Health and Social Care. BB is also supported by the NIHR Applied Research Collaboration North West Coast (ARC NWC). GO is supported by the NIHR School for Public Health Research. IB is supported by NIHR Senior Investigator award.

**Patient and public involvement** Patients and/or the public were not involved in the design, or conduct, or reporting, or dissemination plans of this research.

**Patient consent for publication** Not applicable.

**Ethics approval** University Research Ethics Committee review is not required by the University of Liverpool for the secondary analysis of data that have been anonymised by an external party and are provided to the research team in a fully anonymised format.

**ORCID iDs**
Xingna Zhang http://orcid.org/0000-0002-8849-2112
Gwilym Owen http://orcid.org/0000-0002-5258-8972
Mark A Green http://orcid.org/0000-0002-0942-6628
Iain Buchan http://orcid.org/0000-0003-3392-1650
Ben Barr http://orcid.org/0000-0002-4208-9475

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
