## [Reviewer comments · BMJ Open]

ARTICLE DETAILS

TITLE (PROVISIONAL)	Evaluating the impacts of tiered restrictions introduced in England, during October and December 2020 on COVID-19 cases: A synthetic control study
AUTHORS	Zhang, Xingna; Owen, Gwilym; Green, Mark; Buchan, Iain; Barr, Ben

VERSION 1 – REVIEW

REVIEWER	Jennifer McKinley Queen's University Belfast, School of Natural and Built Environment
REVIEW RETURNED	25-Aug-2021

GENERAL COMMENTS	Review bmjopen_2021-054101 This is an interesting and timely paper and I recommend publication following some consideration of the points below: Abstract: Conclusions what sort of inequalities? social deprivation? This would need to be explained as all forms of inequality were not investigated Strengths and limitations 'We were only able to use data on small neighbourhood areas, rather than on individuals and therefore were unable to investigate how effects of control measures varied by individual or household characteristics – e.g. ethnicity, occupation or household size'. Could this characteristics on ethnicity, occupation or household size for Small neighbourhood areas or LAs be inferred from census information for LAs? Line 58 'We believe our study is also the first to investigate whether such an impact differs by socioeconomic deprivation.' Although not testing the latest tier restriction a useful comparative paper which looked at socioeconomic deprivation impact during the first period of societal lock down is McKinley J.M, Cutting D, Anderson N, et al. Association between community-based self-reported COVID-19 symptoms and social deprivation explored using symptom tracker apps: a repeated cross-sectional study in Northern Ireland. BMJ Open 2021;11:e048333. doi: 10.1136/bmjopen-2020-048333 Introduction Lines 27-36 It is not clear how relevant this is for the current study as the tier restrictions did not impact on working conditions Methods – Data and settings Lines 56-58
---

	Could the method of imputation inflate the numbers for these areas? the number of cases may be higher for the LA than for the MSOAs? Lines 13-19 It isn't clear if the factors mentioned by the authors in addition to the IMD a composite measure of socioeconomic disadvantage, were all analysed individually or as a multivariate sets or form part of the composite IMD measure? Please define how population density was calculated or provide the source of the data The highlights/limitations following the abstract mention that ethnicity could not be accounted for - It is not clear why this is the case if the data for BAME groups was used in the analysis. Please provide more clarity on this. Analysis Please add the year to tables and in the line 42. It would help the flow of the paper to refer to tables and figures within the text rather than used to start sentences. Discussion Line 19-20 'rather than for example employment in other sectors, and involvement in these activities may not differ markedly between socioeconomic groups.' I don't find this sentence very clear. The restrictions did not include a change to employment or working practices but these are markedly different between socioeconomic groups. More deprived areas may be more inclined to work in employment which may not allow/ facilitate working from home. Please revise to provide greater clarity for the reader.
--	---

VERSION 1 – AUTHOR RESPONSE

Reviewer: 1

Dr. Jennifer McKinley, Queen's University Belfast

Comments to the Author:

Review bmjopen_2021-054101

This is an interesting and timely paper and I recommend publication following some consideration of the points below:

Author response: Thank you!

Abstract: Conclusions

what sort of inequalities? social deprivation?

This would need to be explained as all forms of inequality were not investigated

Author response: Thank you for pointing this out. We have clarified it as “socioeconomic inequality”.

The revised sentence now reads as follows on Page 2:

“Compared to Tier 2 restrictions, additional restrictions in Tier 3 areas in England had a moderate effect on transmission, which did not appear to increase socioeconomic inequalities in COVID-19 cases.”

Strengths and limitations

'We were only able to use data on small neighbourhood areas, rather than on individuals and therefore were unable to investigate how effects of control measures varied by individual or household characteristics – e.g. ethnicity, occupation or household size'.

Could this characteristics on ethnicity, occupation or household size for Small neighbourhood areas or LAs be inferred from census information for LAs?

Author response: Thank you for this suggestion. It would have been interesting to explore this aspect.

We have included the proportion of the Black Asian and Minority Ethnic (BAME) groups and the

proportion of students living in an area, population density etc in the matching as potential confounders. We just utilise the indices of deprivation to look at differences in effect related to area based socioeconomic conditions as this includes a combination of multiple measures, including housing overcrowding for example. These are all highly correlated with other areas-based measures such as occupation and the proportion of people from black and ethnic minority groups. It is therefore not possible to disentangle any separate interactions between these different area-based measures, given that they are highly correlated with each other and the limited granularity of the data available. As there may be differential effects at both the area and individual level – individual level data would be needed to understand these potential interactions.

Line 58

‘We believe our study is also the first to investigate whether such an impact differs by socioeconomic deprivation.’

Although not testing the latest tier restriction a useful comparative paper which looked at socioeconomic deprivation impact during the first period of societal lock down is

McKinley J.M, Cutting D, Anderson N, et al. Association between community-based self-reported COVID-19 symptoms and social deprivation explored using symptom tracker apps: a repeated cross-sectional study in Northern Ireland. *BMJ Open* 2021;11:e048333. doi: 10.1136/bmjopen-2020-048333
Author response: We think this is an excellent suggestion. We have added reference (No.7) of this paper on Page 4 in the revised manuscript in the revised manuscript.

We have also revised the ‘Strengths and limitations’ section on Page 2-3.

Introduction

Lines 27-36

It is not clear how relevant this is for the current study as the tier restrictions did not impact on working conditions

Author response: We thank the reviewer for pointing this out. We have further clarified it by rewording as follows - “This is because people of disadvantaged communities may not have the same access to outdoor spaces for socialising, there may be differences in the use of restaurants and pubs and differences on the effectiveness of communications aiming to increase compliance with restrictions”

Methods – Data and settings

Lines 56-58

Could the method of imputation inflate the numbers for these areas? the number of cases may be higher for the LA than for the MSOAs?

Author response:

No, the method of imputation will not inflate the numbers. For example, we may have missing data for 3 MSOAs, but we know the total cases for the non-missing MSOAs (95 cases) and the total for the LA as a whole (100 cases). We therefore know that the difference is 5 cases and that they were in the 3 MSOAs with missing data. We thus impute 1.67 cases for each of the three missing MSOAs – so the total for the LA remains the same. So on average the number of cases in the missing cases will not be inflated. We have also added summary statistics of the number of cases at both MSOA and LA levels before and after imputation in Appendix 1 to show this better.

Lines 13-19

It isn't clear if the factors mentioned by the authors in addition to the IMD a composite measure of socioeconomic disadvantage, were all analysed individually or as a multivariate sets or form part of the composite IMD measure?

Author response: We use the indices of multiple deprivation at two point in the analysis. Firstly, in the matching as a potential confounder and secondly in the interaction analysis to investigate differential effects by level of deprivation. In both cases we use the IMD composite score, which is a weighted combination of multiple domains. The other measures we use in constructing the synthetic controls are average number of care home beds per capita from the Care Quality Commission, population density, the percentage of the over 70 and 7-11 population using 2019's mid-year population estimates from the Office for National Statistics, the proportion of the Black Asian and Minority Ethnic (BAME) groups. These are all included as separate variables and are not part of the IMD. These

factors were all analysed as a multivariate set. We have also revised the manuscript to make it clearer.

The revised text on Page 5 now reads as follows:

“These included the overall score of the English Indices of Multiple Deprivation (IMD) - a composite measure of socioeconomic disadvantage that combines 7 domains of deprivation (income, employment, education, health, crime, barriers to housing & services, and living environment)”.

Please define how population density was calculated or provide the source of the data

Author response:

These are the mid-2019 population estimates for each MSOA divided by area of the MSOA in hectares from the Office for national statistics -

<https://www.ons.gov.uk/peoplepopulationandcommunity/populationandmigration/populationestimates/bulletins/annualsmallareapopulationestimates/mid2019/relateddata>

<https://geoportal.statistics.gov.uk/datasets/middle-layer-super-output-areas-december-2001-boundaries-ew-bfc/explore?location=52.950000%2C-2.000000%2C6.65>

We have added to the text outlining this and referencing the data source on Page 5.

The highlights/limitations following the abstract mention that ethnicity could not be accounted for - It is not clear why this is the case if the data for BAME groups was used in the analysis. Please provide more clarity on this.

Author response: We meant the individual-level ethnic characteristics couldn't be accounted for in our study, as we relied on aggregated area-based measures in our analysis (the proportion of the population from a BAME group) and we don't have access to individual-level ethnicity data. We have revised the 'Strengths and limitations' section following the abstract to clarify that.

Analysis

Please add the year to tables and in the line 42.

Author response: Thank you for pointing this out. The reviewer is correct, and we have added the year 2020 to the main text in Line 48 on Page 5, Table 1 on Page 7, and Table 2 on Page 8 accordingly.

It would help the flow of the paper to refer to tables and figures within the text rather than used to start sentences.

Author response: As suggested by the reviewer, we have revised the text throughout the manuscript in Line 3, Line 43, and Line 56-57 on Page 7, and Line 10-11 on Page 8.

Discussion

Line 19-20

'rather than for example employment in other sectors, and involvement in these activities may not differ markedly between socioeconomic groups.'

I don't find this sentence very clear. The restrictions did not include a change to employment or working practices but these are markedly different between socioeconomic groups. More deprived areas may be more inclined to work in employment which may not allow/ facilitate working from home. Please revise to provide greater clarity for the reader.

Author response:

Thank you, this was not clear enough. What we intended to say was that, as the reviewer points out, – because the restrictions did not require people to work from home and capacity to work from home differs markedly between socioeconomic groups, that might explain why we did not see differential effects by socioeconomic group, whilst this might be expected from broader lockdowns. We have rephrased this sentence to make this clearer.

The revised text now reads “Whilst broader lockdowns that require people to work from home where possible – might be expected to have differential effects by socioeconomic group, with more disadvantaged groups being less likely to be able to work from home, this is not necessarily the case with restrictions to the hospitality sector. It may be the case that transmission within the hospitality sector occurs at a similar level in more deprived and more affluent neighbourhoods and the restrictions introduced reduce these risks by similar amounts.”

VERSION 2 – REVIEW

REVIEWER	Jennifer McKinley Queen's University Belfast, School of Natural and Built Environment
REVIEW RETURNED	15-Mar-2022
GENERAL COMMENTS	The authors have addressed all of the review comments. I look forward to seeing this research published.